# The Influence of Metabolism on Immune Response: A Journey to Understand Immunometabolism in the Context of Viral Infection

**DOI:** 10.3390/v15122399

**Published:** 2023-12-09

**Authors:** Daed El Safadi, Aurélie Paulo-Ramos, Mathilde Hoareau, Marjolaine Roche, Pascale Krejbich-Trotot, Wildriss Viranaicken, Grégorie Lebeau

**Affiliations:** 1PIMIT—Processus Infectieux en Milieu Insulaire Tropical, Université de La Réunion, INSERM UMR 1187, CNRS 9192, IRD 249, Plateforme CYROI, 97490 Sainte-Clotilde, France; daed.el-safadi@univ-reunion.fr (D.E.S.); marjolaine.roche@univ-reunion.fr (M.R.); pascale.krejbich@univ-reunion.fr (P.K.-T.); 2INSERM, UMR 1188 Diabète Athérothrombose Réunion Océan Indien (DéTROI), Université de La Réunion, Campus Santé de Terre Sainte, 97410 Saint-Pierre, France; aurelie.paulo-ramos@univ-reunion.fr (A.P.-R.);

**Keywords:** antiviral response, immunometabolism, viral infections, metabolic diseases

## Abstract

In recent years, the emergence of the concept of immunometabolism has shed light on the pivotal role that cellular metabolism plays in both the activation of immune cells and the development of immune programs. The antiviral response, a widely distributed defense mechanism used by infected cells, serves to not only control infections but also to attenuate their deleterious effects. The exploration of the role of metabolism in orchestrating the antiviral response represents a burgeoning area of research, especially considering the escalating incidence of viral outbreaks coupled with the increasing prevalence of metabolic diseases. Here, we present a review of current knowledge regarding immunometabolism and the antiviral response during viral infections. Initially, we delve into the concept of immunometabolism by examining its application in the field of cancer—a domain that has long spearheaded inquiries into this fascinating intersection of disciplines. Subsequently, we explore examples of immune cells whose activation is intricately regulated by metabolic processes. Progressing with a systematic and cellular approach, our aim is to unravel the potential role of metabolism in antiviral defense, placing significant emphasis on the innate and canonical interferon response.

## 1. Introduction

Metabolism is generally referred to as the sum of all biochemical reactions within an organism designed to produce or consume energy and, by extension, allow the maintenance of life. Metabolism thus includes all the anabolic or catabolic reactions occurring at the cellular or systemic level. On the one hand, it provides the energy necessary for the cell to ensure its activities, whether in the form of ATP or reduced substrates such as NADH or NADPH. On the other hand, via anabolic pathways, metabolism participates in the synthesis of biomass (nucleic acids, amino acids and proteins, lipids, and carbohydrates) useful for the storage of energy and information, or for the formation of functional units of the cell (i.e., proteins). The balance between anabolism and catabolism is finely regulated to preserve energy homeostasis. This regulation is dependent on enzymes that are very sensitive to changes in energy status (i.e., glucose deprivation vs. high glucose level). Indeed, the cell seeks to maintain an adequate ATP/ADP ratio with either cellular functions or growth.

As metabolism is the keystone of cellular activity, the mechanisms through which it is governed deserve attention. It should be noted that the consequences of metabolic disorders on cell functions are studied in the specific case of human pathologies, such as diabetes, obesity, or cancer. Immune responses (i.e., innate and adaptive) as well as pathogen sensing are among the issues for which a close link with metabolism has also been established [1]. During viral infections, particular metabolic states, namely the metabolic pathways active at the time of infection, seem to be favorable to virus multiplication. This may be through a direct action of some metabolites in the early stages of the viral cycle or the establishment of antiviral responses (notably interferon-dependent) [2].

To maintain its homeostasis in a fluctuating environment, the cell must have a consistent metabolic status so that a shift in the dynamics of cell metabolism is possible depending on the cell’s fate. Schematically, a quiescent cell will tend to favor the use of carbohydrate substrates, degraded to pyruvate to fuel the tricarboxylic acid cycle (TCA), and provide large amounts of energy in the form of ATP. This mechanism is driven via oxidative phosphorylation (OXPHOS), which enables cells to maintain their activities (e.g., enzymatic activities, ATP-dependent pump, ATP/ADP ratio-dependent signaling). By contrast, a proliferating cell will favor the rapid production of energy through ‘aerobic glycolysis’, leading to the utilization of pyruvate to produce lactate even under aerobic conditions (Figure 1). This mechanism, known as the Warburg effect, was initially observed in cancer cells and tends to facilitate the abundant synthesis of biomass at the expense of efficient energy production [3]. More recently, the ability of some viruses to modulate the metabolism in this way has been established, reminiscent of the Warburg effect [4,5]. During infection, the virus exploits the cellular machinery to its advantage, aiming to rapidly produce and release an important quantity of virions. This intense viral replication within the infected cell necessitates a significant amount of nucleotides, amino acids, and lipids, all supplied in a manner similar to that in a cancer cell, through metabolic reprogramming [4]. This ability to reprogram or redirect the host’s metabolism seems to be shared by many viruses, spanning different families and groups, such as the human immunodeficiency virus (HIV), dengue virus (DENV), severe acute respiratory syndrome coronavirus 2 (SARS-CoV-2), or herpes simplex virus (HSV) [4,6]. Moreover, it is plausible that the tendency of viruses to redirect metabolism could elucidate the preferential targeting of cells engaged in aerobic glycolysis, such as tumor cells. These cells, with an existing metabolism suitable for infection and viral replication, emerge as a prime target. This preference may confer oncolytic potential to the infecting virus, capable of reinstating immune responses within tumors.

Another putative reason for modulating the host’s metabolism during infection is to impede the establishment of the antiviral cellular response, thereby promoting viral expansion. The hypothesis that certain transient or specific metabolic states due to chronic diseases favor viral infection takes on full significance in this context, partly explaining the differences in systemic reactivity to the same virus.

Indeed, the interactions between metabolism and immunity are increasingly discussed with the emergence of the term ‘immunometabolism’. This facet of immunology has been the subject of numerous advances in the context of cancer (see Section 2.1
*Immunometabolism and cancer*) and has already been extensively presented elsewhere. The aim of this review is rather to discuss, in light of the latest available literature, how metabolic disorders (diabetes, obesity, and others) might affect the outcome of viral infections. Then, we will discuss the advances made regarding the interplays between metabolism and immunity in the context of viral infections, with an emphasis on the antiviral innate immunity of the infected host cell.

## 2. Insight into Immunometabolism Concept

The immune system is an intricate network of cells and molecules that safeguard the body against threats. Recently, increasing attention has been directed toward the regulation of this network through metabolism, giving rise to the concept of immunometabolism. The study of immunometabolism could provide insight into how obesity fosters inflammation in conditions such as arthritis [7]. It also offers a way of understanding how drugs act on metabolism or metabolites, such as methotrexate or 25-hydroxycholesterol, to find utility in the treatment of infectious diseases [8,9].

### 2.1. Immunometabolism and Cancer

A proper introduction to immunometabolism is its contribution to the understanding and treatment of pathologies of cancer origin. Indeed, cancer has been one of the most prolific subjects of study for immunometabolism. Cancer is a group of pathologies that can occur in virtually all organs and have in common the anarchic division of cells, possibly leading in the case of malignant tumors to the invasion of distant tissues. The metabolism of cancer cells is of crucial importance for the development of these pathologies, as cancer cells have a particular type of metabolism that allows them to support their development. In addition, the tumor metabolic environment also influences the immune system by participating in immune tolerance and immune system failures in the clearance of tumor cells [10].

Indeed, tumors have been shown to modulate the concentration of nutrients critical to immune cell function. The infiltration of immune cells into the tumor tissue is generally associated with a better prognosis [11]. In particular, T cell infiltration is well documented, and the promotion of their activity is even the subject of new therapeutic approaches [12]. The tumor cells’ microenvironment is specific and results from the metabolism of tumor cells. The latter is based primarily on the high rate of glycolysis to allow the large production of biomass (e.g., amino acids, nucleotides, and lipids) required for their sustained division. This leads to the depletion of the available intratumoral glucose pool [13]. Notably, glucose depletion has been associated with decreased infiltration and reactivity of the immune system, including T cells at the tumor site [14]. For example, a deficit in the production of IFN-γ, a key effector secreted by tumor-infiltrating CD8^+^ T cells, was observed after glucose deprivation [15]. This decrease is induced by the binding of glyceraldehyde phosphate dehydrogenase (GAPDH) to the 3′ untranslated region (3′UTR) of the IFN-γ mRNA when the glycolytic enzyme is not devoted to its metabolic function, namely, at low levels of glucose [15]. Alternatively, phosphoenolpyruvate (PEP), an intermediate glycolysis metabolite, whose level decrease following glucose deprivation, is essential for T cell antitumor responses [16]. It supports T cell receptor (TCR) and nuclear factor of activated T cell (NFAT)-induced signaling, whose activation depends on inositol trisphosphate receptors (IP3Rs) located on the mitochondrial membrane [16]. Ca^2+^ then acts as an integrator of TCR signaling and PEP by inhibiting sarco/ER Ca^2+^−ATPase (SERCA) activity. This increased cytosolic Ca^2+^ concentration then promotes the antitumor activities of TCR-dependent CD4^+^ and CD8^+^ T cells [16].

### 2.2. A Case of Metabolic-Dependent Activation in Immune Cells: Monocytes and Macrophages

Macrophages are CD11b^+^CD68^+^ leukocytes historically described for their antimicrobial and phagocytic activities. However, many other functions have been associated with them over the years [17]. Indeed, their participation in tissue homeostasis, as well as tissue repair in case of damage, is no longer to be proven [17]. To fulfill these essential but diverse functions, macrophages can adopt a variety of phenotypes. Given the complexity of these phenotypes, it is generally accepted to classify macrophages according to whether they have proinflammatory (M1) or anti-inflammatory (M2) activities [17]. Nevertheless, it is important to keep in mind that this classification greatly simplifies the repolarization capabilities of these cells, which rely on transcriptionally dynamic capacities to adopt the phenotype and functions best suited to their tissue microenvironment [17]. Moreover, macrophages are among the only cells to have so many tissue-specific counterparts (i.e., Kupffer cells in the liver, microglia in the brain, and alveolar macrophages in the lung) [17]. All these tissue-specific macrophages have an embryonic origin, unlike the monocyte lineage—macrophages, which derive from the differentiation of monocytes following their tissue infiltration. In what follows, we will discuss the metabolic characteristics of macrophages according to whether they are M1- or M2-polarized in vitro and in vivo. Some differences that may occur depending on their tissue of residence have been reviewed very recently by Wculek et al. [18].

M1 macrophages (CD11b^+^CD68^+^CD64^+^) have a predominantly proinflammatory activity whose activation in vitro occurs either through TLR stimulation via LPS or cytokine stimulation with TNF or IFNγ. This activation reprograms cell metabolism in a way that supports the establishment of an oxidative burst through the production of reactive oxygen species (ROS) and nitrogen species (RNS), as well as the production of antimicrobial effectors. M1 macrophages are characterized by predominant glycolytic metabolism (Figure 2), as evidenced by their increased consumption of glucose, enabled in particular by an increase in the glucose type 1 receptor (GLUT1) [19,20]. Hypoxia-inducible factor-1α (HIF-1α) was shown to be a transcription factor essential for the onset of this metabolic reprogramming [21,22]. The stabilization of HIF-1α is mediated in M1 macrophages by blocking TCA at two distinct levels. First, the inhibition of isocitrate dehydrogenase (IDH-1) leads to citrate accumulation, as well as itaconate synthesis [23]. On the one hand, the accumulated citrate is converted to acetyl-CoA, which is used for histone acetylation and inflammatory gene transcription [24]. On the other hand, itaconate inhibits the activity of the succinate dehydrogenase complex (SDH), leading to the accumulation of succinate [25]. Ultimately, the succinate generated in this way stabilizes HIF-1α [22]. Notably, this metabolic reprogramming has been shown to be essential for IL-1β production in response to LPS stimulation [22]. Mitochondrial activity in M1 macrophages is then diverted from ATP production to mitochondrial ROS production. The latter will serve within phagosomes, participating in antigen processing and presentation [26]. At this stage, ATP production is therefore mainly dependent on aerobic glycolysis, similar to the effect described by Warburg for cancer cells [5,22]. This also supplies the necessary intermediates for the pentose phosphate pathway (PPP), leading in the end to an increase in the NADPH pool, as well as to the production of biomass (here as nucleotides) and ROS [27].

Unlike M1 macrophages, macrophages with a M2 phenotype (CD11b^+^CD68^+^CD163^+^) are obtained in vitro in response to IL-4 or IL-13 and display anti-inflammatory properties as well as involvement in tissue repair [17]. In contrast to M1, M2 macrophages have a cellular metabolism based on increased OXPHOS and mitochondrial respiration (Figure 3), as evidenced by the elevation of OCR in macrophages polarized toward an anti-inflammatory phenotype [28]. Transcriptional shifts leading to this metabolism are controlled in M2 macrophages through the activation of several factors, including PPARδ, PPARγ, PGC-1β, and IRF4. The activation of these factors depends on both STAT6 and the mTORC2 complex [28,29]. Some of these factors are associated with lipid metabolism and lead, particularly here, to the upregulation of the scavenger membrane protein CD36 that allows the internalization of lipids used for fatty acid oxidation (FAO) [29,30]. While CD36 has been shown to be critical for polarization in M2 [30], the role of FAO remains ambiguous. Indeed, the use of etoxomir to inhibit FAO or genetic models that prevent it do not appear to affect M2 polarization [31,32], but rather it is the ability of M2 macrophages to produce ATP that affects this process. Thus, one might conclude that FAO does not appear to be essential for M2 macrophages to perform their inflammatory functions, with fatty acids serving instead as a substrate for energy production through OXPHOS [31]. In contrast, glutaminergic metabolism appears to be of paramount importance with glutamine providing one-third of the carbon derivatives of TCA [33]. Glutamine leads to the accumulation of α-ketoglutarate, which orchestrates M2 polarization by activating Jumonji domain-containing protein D3 (JMJD3) demethylases [34]. They mediate the histone demethylation of M2-specific gene promoters [34]. As a result, these epigenetic modifications are partly responsible for M2 polarization.

As with other cells of the immune system, deficits in macrophage metabolism can occur pathologically, as in the case of atheromatous intraplaque foamy macrophages or in obesity [18].

### 2.3. The Metabolic-Dependent Activation of Tissue Sentinels: Dendritic Cells

Dendritic cells (DCs, CD11c^+^) are antigen-presenting cells (APCs), which like macrophages, have variable origins [35]. In fact, several subtypes of DCs can be distinguished according to their origin. Conventional DCs (cDC1 and cDC2) and plasmacytoid DCs (pDCs) are derived from specific progenitors, and others derived from monocytes (mo-DCs) [35]. As APCs, DCs are found in a resting state in many tissues (e.g., Langerhans cells in the dermis) where their role is to sense their environment and then activate in case of danger. Activation occurs in response to pathogen-associated molecular patterns (PAMPs) or damage-associated molecular patterns (DAMPs) and results in initiating the immune response through the secretion of cytokines, as well as the presentation of antigens in lymphoid tissues [35].

The metabolism of resting DCs has already been studied both in vitro and in vivo. Thus, it is known that the mTORC1 complex has a preponderant activity here again. Indeed, in mice, the use of rapamycin leads to the inhibition of the ex vivo growth of pDCs and cDCs from the bone marrow (BM); in vivo, the pool of DCs is considerably reduced [36]. Confirming these results obtained in mouse models, the use of rapamycin on human mo-DCs results in impaired survival [37]. Conditional expression models were used to further explore the importance of mTORC1 in DC survival. Phosphatase and TENsin homolog (*Pten*) deletion using a *Cd11c*-Cre system leads to the expansion of DCs in mice. This result is consistent with the previous one, PTEN being a negative regulator of mTOR [36]. The deletion of Tsc1, another negative regulator of mTOR, using a similar system, led to an increased growth of cultured BM-DCs [38]. As one of the main factors that regulate cellular metabolism, mTORC1 directs quiescent DCs to use OXPHOS to satisfy their low energy requirement [39].

As for the other immune cells presented so far, the activation of DCs following the stimulation of one of their pattern recognition receptors (PRRs) leads to a shift in metabolism, with glucose becoming the main fuel for cellular metabolism (Figure 4). This process involves two steps. Indeed, shortly after the activation of DCs in a TLR-dependent manner, a rapid increase in glycolytic activity and the entry of intermediates into the PPP was observed [40]. These changes are initiated through the activation of AKT via TANK-binding kinase 1 (TBK1) and IκB kinase ε (IKKε) [40]. AKT phosphorylates and subsequently activates hexokinase 2 (HK2), the glycolysis-limiting enzyme whose activity increases after association with the mitochondrial surface [40]. Lately, the maintenance of glycolysis within DCs relies on activation by the mTOR complex of HIF-1α [41]. Interestingly, if the inhibition of HIF-1α or mTOR occurs early after TLR stimulation, it has no impact on DC activation, in contrast to the early inhibition of AKT. This illustrates the importance of the timing of different signaling pathways in DC activation [40]. In contrast, the activation of the AMP-activated protein kinase (AMPK) pathway appears to control DC activation by promoting OXPHOS, in particular, through PGC1α activation, which favors mitochondrial biogenesis. In fact, AMPK knockdown has been associated with a potentiation of DC activation [42], whereas its activation or that of PGC1α leads to tolerogenic DCs [42].

Therefore, we have shown here that essential changes in immune cell metabolism are required during their activation to enable them to support their function, whether as a support or effector. We will now discuss the contributions of immunometabolism in certain pathological situations.

## 3. At the Systemic Level: Interplays between Host Metabolic Status and Infectious Diseases

It is gaining increasing awareness that the resolution of an infection depends on multiple intrinsic physiological factors. In this light, the host metabolic status and the related comorbidities are now taken into account in understanding immune responses, which are often more or less efficient during the course of an infectious disease, conditioning resolution, or subsequent pathophysiological processes.

### 3.1. Diabetes, Obesity, and Other Metabolic Disorders

Metabolic diseases (e.g., type 2 diabetes, obesity, etc.) represent a growing risk for the world’s population, particularly through their cardiovascular (myocardial infarction, stroke, etc.) and neurological (diabetic neuropathy) complications. Previously restricted to high-income countries and Western lifestyles, these diseases are now tending to develop in low- and middle-income countries. As depicted in this section, these metabolic disorders may present an associated risk of infection. It was also recently demonstrated during the COVID-19 epidemic that one of the complications of SARS-CoV-2 infection is the development of post-infection diabetes.

#### 3.1.1. Impact of Diabetes on Infectious Diseases

Diabetes is a chronic disease characterized by hyperglycemia and chronic low-grade inflammation. Two main types of diabetes can be distinguished. Type 1 or insulin-dependent diabetes (T1D) is an autoimmune disease targeting the β-cells of the islets of Langerhans in the pancreas which secrete insulin, a key hormone in the regulation of circulating glucose. In contrast, the pathophysiology of type 2 or insulin-resistant diabetes (T2D) involves hyperinsulinemia, due to the progressive resistance of cells to insulin following the installation of a low-grade inflammation [1]. Diabetes accounts for a total of more than 460 million cases in 2019 with a projection to 700 million cases in 2045, the majority of cases being T2D [43]. Beyond its cardiovascular and neurological complications, diabetes could lead to impaired host defenses and an increased risk of infection.

In fact, diabetes has been found to cause immune deficiencies in several essential compartments to establish an adequate immune response. Chronic hyperglycemia is accompanied by the production of advanced glycation end products and ROS. This pro-oxidant status promotes chronic low-grade inflammation, which is detrimental to the immune response. More specifically, T2D is accompanied by the recruitment of M1 polarized macrophages within the adipose tissue where they will secrete significant amounts of proinflammatory mediators (tumor necrosis factor α—TNF-α, C-reactive protein—CRP, interleukin-1β—IL-1β, CCL2, CCL3, CXCL8, IL-6, and IL-12) [44,45]. However, under bacterial stimulation, peripheral blood mononuclear cells (PBMCs) from diabetic patients have been found to have a decrease in functional capabilities compared to healthy individuals. Thus, a decrease in the secretion of cytokines such as IL-1β, IL-6, and IL-2 is observed [46,47,48]. Notably, IL-6 plays an important role in protection against pathogens and in the adaptive immune response by inducing antibody production and the development of effector T cells [49]. Therefore, these studies reveal that the inhibition of these cytokines in hyperglycemia can affect the immune response development against pathogens. In addition, the suppression of type I interferon (IFN) production is reported in PBMCs cultured under high glucose conditions and stimulated with viral RNA mimetic, namely polyinosinic–polycytidylic acid (poly:IC) [50]. It was also demonstrated that leukocyte recruitment is inhibited in a diabetic context, notably due to an alteration in the expression of adhesion molecules [51]. More specifically, for macrophages, there is a decrease in Fcγ receptor expression, as well as in the antimicrobial and phagocytosis capacities [52,53]. Similarly, for neutrophils, a reduction in antimicrobial capacities was noted. This is due to an alteration in the pro-oxidant and degranulation capacities [54,55,56]. Furthermore, the production of neutrophil extracellular traps (NETs) or NETose is decreased under hyperglycemic conditions, leading to increased susceptibility to infections [57]. Deficiencies in degranulation have also been demonstrated in NK cells from T2D subjects, resulting from a defect in NKG2D- and NKp46-activating receptors [58]. Notably, complement activation is impacted in hyperglycemia, due to the dysfunction of opsonization through the C4 fragments of complement [59].

In addition to effector cells, T2D is accompanied by an altered presentation and thus pathogen detection. Therefore, diabetic mice showed a reduced expression of the adaptor protein containing the Toll-like receptor 2 (TLR) and the Toll/IL-1R domain (TIRAP) [60], involved in the detection of pathogens. However, contradictory data exist since it has been shown that the expression of TLRs by monocytes and neutrophils from diabetic subjects was increased [61]. In fact, it is now known that the quality of blood glucose control has an impact on TLR expression. A subject with hyperglycemia and poor control will have decreased TLR expression [61], while a subject with controlled hyperglycemia will have higher TLR expression [61]. Dendritic cells, the main antigen-presenting cells (APCs), showed delayed recruitment to the site of infection in diabetic mice in the course of *M. tuberculosis* infection [62]. This delay could be attributed, as previously for monocytes–macrophages, to reduced levels of CCL2 and CCL5 [62]. In addition, in the context of intracellular bacterial infection, this delay in recruitment has been shown to be accompanied by impaired phagocytosis [63].

Diabetes has been repeatedly shown to be a risk factor for developing bacterial and fungal infections, and this has already been reviewed [64,65]. However, viral infections have less been reported. Nevertheless, a study including nearly 190,000 people in China showed that diabetic individuals have a 1.5 times higher relative risk of developing hepatitis B virus infection than nondiabetic individuals [66]. Associated with the immunological disorders discussed above, Middle East respiratory syndrome coronavirus (MERS-CoV) infection in diabetic mice is accompanied by unresolved lung inflammation, leading to exacerbated pathology [67]. These mice were also shown to have fewer monocytes–macrophages and CD4^+^ T cells, without affecting viral clearance [67]. For SARS-CoV-2, chronic inflammation found in T2D appears to be a risk factor for developing severe forms of COVID-19 [68]. Furthermore, strengthening the link between TD2 and viral infection, it is now known that SARS-CoV-2 can infect adipocytes, leading to a decrease in adiponectin secretion. This contributes to the establishment of an inflammatory state conducive to the development of insulin resistance [69], explaining the increased risk of transitioning from a prediabetic state to diabetes in COVID-19 patients. Finally, the complications of diabetes include cardiovascular damage (atherothrombosis, stroke, myocardial infarction, etc.). Many of these cardiovascular complications are related to endothelial damage. Moreover, severe forms of DENV are known to be associated with damage to the endothelial barrier. Therefore, it is hypothesized in the literature that severe forms of DENV infection are more likely to occur in diabetic individuals due to pre-existing endothelial damage [70,71].

#### 3.1.2. Impact of Obesity on Infectious Diseases

Overweight and obesity are defined as an excessive accumulation of fat in an individual. The abnormal accumulation of visceral adipose tissue (VAT) is a risk factor for developing cardiovascular events and is often associated with the development of insulin resistance [72]. Previously limited to high-income countries, overweight and obesity are now also increasing rapidly in low- and middle-income countries. According to the WHO, today, more people are suffering from obesity than from malnutrition [73]. In 2016, there were 1.9 billion overweight people, over 650 million of whom were obese [73].

Obesity is a multifactorial metabolic disorder combining genetic, environmental, and developmental factors [74], whose simplest cause corresponds to a caloric intake that is not consistent with the individual’s energy expenditure. However, more complex mechanisms can be underlined, such as impairment in the neurons of the arcuate nucleus of the hypothalamus involved in the control of food intake. This is mediated by the expression of orexigenic factors (neuropeptide Y, agouti-related protein, etc.) or anorexigenic factors (pro-opiomelanocortin, cocaine- and amphetamine-regulated transcript—CART, etc.) under the hormonal control of leptin and insulin secreted by adipocytes and β-cells of the islets of Langerhans, respectively [74]. More recently, the impact of the dysbiosis of the intestinal microbiota on the risk of obesity has been discussed. Indeed, the abundance of firmicutes bacteria seems to be linked to the development of obesity, due to the ability of these bacteria to extract energy more efficiently, thus promoting calorie absorption and weight gain [75].

Beyond the cardiovascular and endocrine complications associated with obesity, obese individuals exhibit immune disorders. In addition to its important storage role, VAT is a tissue that is strongly linked to immunity, in particular through the secretion of adipokines (leptin and adiponectin), cytokines (TNF-α and lL-6), and chemokines (CCL2). They exhibit activity not only in the inflammatory state but also in the immune system [76]. Thus, hyperplasia and hypertrophy in obesity are accompanied by the deregulation of the immune system, which we will describe here. Adipose tissue contains different cell types besides adipocytes, including endothelial cells, fibroblasts, pre-adipocytes, and especially leukocytes [77]. First, in obese subjects, there is an alteration in the recruitment, proliferation, and polarization of macrophages within the adipose tissue [78]. Increased adipocyte size has been shown to trigger a stress response and the release of chemokines (CCL2 and CCL5) [79], leading to the recruitment and accumulation of monocytes and macrophages in adipose tissue [78]. On the other hand, the increased presence of free and nonesterified fatty acids in obesity results in the activation of TLR4 and TLR2, leading to the production of proinflammatory cytokines (e.g., IL-6) and reduced expression of anti-inflammatory cytokines (e.g., IL-10) [80,81]. This proinflammatory state may ultimately be responsible for the development of insulin resistance and T2D, as previously explained [81]. Finally, while most M2 polarized macrophages are found in healthy subjects’ adipose tissue, macrophage polarization in obese individuals is oriented toward the M1 phenotype [77]. This accumulation is, on the one hand, due to the recruitment of blood monocytes, but also the proliferation and polarization shift of resident macrophages in adipose tissue [82]. Leukocytosis in adipose tissue in obesity also involves neutrophils. Similar to monocytes, neutrophils from obese subjects show increased migration within adipose tissue, as well as increased basal superoxide production, thus participating in low-grade inflammation [83]. Finally, NK cells follow this trend, with increased activation, proliferation, and IFN-γ secretion [84]. Cells involved in adaptive immunity are also not spared in obesity. Indeed, adipose tissue from obese subjects showed a higher presence of CD4^+^ helper 1 and CD8^+^ cytotoxic T cells [85]. Although this proinflammatory status is the one found in the case of infection, its chronicity is accompanied by an enhanced susceptibility to infections and a lower reactivity to pathogens in obese subjects [83,86].

The immune alterations in obesity are responsible for a heightened risk of developing infections. These infections often occur opportunistically or nosocomially in tissues harboring a commensal microbial flora, at the interface between the internal and external medium. Consequently, obese mice have shown insufficient antimicrobial activity to combat *Staphylococcus aureus* during a cutaneous infection [87]. Fungal infections with *Candida albicans* are also more prevalent in obese individuals [88]. Furthermore, there is an increased risk of developing *Escherichia coli* infections in the urinary tract of these subjects [89]. However, it is in the respiratory system that the most data have been reported. Indeed, obese subjects are at a higher risk of contracting upper or lower respiratory system infections [90]. Mice^ob/ob^ have shown alterations in their immune response when exposed to pathogens of bacterial origin, including *Mycobacterium tuberculosis*, *Listeria monocytogenes*, and *Klebsiella pneumoniae* [91,92,93]. Due to the poor control of these infections by the immune system, they are more likely to breach the endothelial barrier, leading to septicemia [91]. The inability to control infection under these conditions has been partly attributed to defects in antigen-specific T cell immune responses [93], as well as deficiencies in the phagocytic activity of neutrophils and macrophages [92]. The severity of infections in obese subjects is also heightened in the case of viral infection. Consequently, the successive epidemics of H1N1 influenza and SARS-CoV-2 have been correlated with an increased risk of mortality and morbidity in obese patients [94,95]. In the case of Influenza virus infection, it has been demonstrated in a murine model that the increased susceptibility of obese individuals is linked to a defect in antigen presentation to T lymphocytes by dendritic cells, leading to deficient cytotoxic and memory responses [96,97]. It is noteworthy that obesity has recently been associated with the greater severity of West Nile virus (WNV) infection in a murine model [98]. This increase in severity was found in females and is linked to the dysfunction of adaptive immunity in the acute phase, as well as in the late phase, with a decrease in the production of neutralizing antibodies in these individuals [98]. Correspondingly, with this inability to produce a humoral response, it is known that obesity has a negative impact on immune response in the case of vaccination. Thus, the immunogenicity induced by vaccines targeting the Hepatitis B and H1N1 Influenza viruses is reduced in obese individuals [99,100].

### 3.2. Controversial Role of Exercise on the Development of Infectious Diseases

Much like nutrition, physical activity has an impact on an individual’s metabolic status; however, its influence on immune function remains a subject of debate (Figure 5). Epidemiological evidence suggests that regular physical activity reduces the incidence of chronic diseases and the risk of viral and bacterial infections [101]. In fact, frequent physical exercise may enhance immune competence. A prospective cohort study of 1509 Swedish men and women aged between 20 and 60 years demonstrated that higher levels of physical activity were associated with a lower incidence of self-reported upper respiratory tract infections [102]. Acute exercise sessions of less than 60 min have been shown to result in a transient alteration in leukocyte counts and lymphocyte proliferation capabilities [103]. Although these changes remained modest, they could be considered as an enhancement of immunosurveillance during moderate physical activity. This type of exercise serves as a crucial adjunct to the immune system, facilitating the continuous turnover of leukocytes between circulation and tissues [104]. Moreover, physical exercise has been suggested as a valuable tool for COVID-19 prevention by improving immune function, particularly innate and adaptive immunity, and modulating the anti-inflammatory state [105].

However, prolonged and intensive exercise in athletes is associated with immune dysfunction, inflammation, and oxidative stress [106,107]. Exercise may transiently impair immune protection, thereby increasing the risk of infection [108]. Consequently, athletes may experience a temporary or sustained reactivation of viruses, such as the Epstein–Barr virus, due to the suppression of immune parameters during years of intense training [109,110,111].

Collectively, these studies demonstrate that physical exercise is beneficial for maintaining functional immunity. However, when practiced at a very high intensity, it is suggested to alter immune function and increase susceptibility to infections (Figure 5). It is also crucial to bear in mind that the connection between exercise and immune function is undoubtedly a multifactorial concept involving other aspects such as nutrition and changes in the microbiome, for example [112]. To this end, further studies using a multifactorial approach will be necessary to enhance our understanding of the immune changes induced by exercise and their underlying mechanisms.

## 4. At the Cellular Level: Metabolism Involvement in Antiviral Innate Response and Sensing

Until now, immunometabolism has been defined as the impact that cellular metabolism may have on the development of the immune response, mainly focusing on cells that are part of the immune system, whether adaptive or innate. Indeed, many reviews focus on immune cells and the metabolic pathways involved in their activation and functionality. However, immunometabolism should also include the early responses established by parenchymal and connective cells after damage or infection. These responses are essential for the immune control of pathogens and the subsequent development of the immune responses themselves. That is why we will now shed light on the cells at the forefront during infection and how their metabolism, and its inherent control during the infection, can contribute to a more or less effective antiviral response.

### 4.1. Metabolic-Dependent Sensing of the Viral Genome and Interferon Secretion

During a viral infection, cells at the infection site develop an antiviral response based on the secretion of type I interferons (including IFN-α and β) and type III interferons (structurally related to the IL-10 family) [113]. In terms of type I interferons, the infected cells will predominantly secrete IFN-β, while IFN-α subtypes concern immune cells and more particularly monocytes and dendritic cells.

The secretion of IFNs and the resulting antiviral response depend on the recognition of viral PAMPs by cellular PRRs. Viral nucleic acids are the most powerful mediators among the PAMPs capable of alerting the infected cell. The PRRs involved in the detection of these viral nucleic acids vary according to their type. The recognition of viral RNA in the cytosol of infected cells is based on the RIG-I-like receptor (RLR) family, which includes RIG-I and MDA5. RIG-I has shown the ability to bind RNA with motifs that are not found in mammals, such as 5′-PPP or 5′-PP caps [114,115]. On the other hand, detection through MDA5 remains to be explored, although MDA5 has been shown to be essential as a sensor for synthetic double-stranded RNA, such as the poly:IC and the picornaviruses [116,117]. In both cases, recognition is followed by a conformational change and oligomerization of RLRs via their caspase activation and recruitment domain (CARD). Subsequently, this domain becomes available to bind to the mitochondrial antiviral signaling protein (MAVS) CARD domain, localized at the external mitochondrial membrane (Figure 6) [118,119]. The RLR–MAVS interaction thus leads to the activation of IFN regulatory factors 3, 7 (IRF3, IRF7) and NF-κB. This activation occurs via a signaling cascade involving TNF3 receptor-associated factor (TRAF3), TANK-binding kinase 1 (TBK1), and IκB kinase-ε (IKKε), resulting in the secretion of type I and III IFNs [120] (Figure 6). The detection of cytosolic DNA is based on the cGAS/STING axis, which, despite the introduction of other sensors, remains the most widely reported pathway in the literature. By detecting cytosolic dsDNA, cyclic GMP–AMP synthase (cGAS) produces cyclic GMP–AMP (cGAMP), leading to the activation of stimulator of interferon genes (STING) [121]. In turn, STING forms a signaling complex with TBK1, inducing IRF3 and the expression of type I and III IFNs (Figure 6) [121]. At the endosomal level, TLRs 3, 7/8, and 9, respectively, detect double-stranded DNA, single-stranded RNA, and DNA containing unmethylated CpGs [122]. Signaling by TLRs leads, via their own adaptor proteins (TIR domain-containing adaptor molecule 1—TICAM1 and myeloid differentiation primary response 88—MYD88), to the same transcriptional factors as for the RLR and cGAS/STING pathways, namely the activation and dimerization of IRFs 3 and 7 [122]. Finally, the viral genome can also be detected via sensors associated with the inflammasome, the nucleotide-binding oligomerization domain (NOD)-like receptors (NLRs), and particularly NLRP3 NOD-like receptor family, pyrin domain-containing 3 (NLRP3). This detection induces MAPK-dependent signaling, associated with the implementation of unspecific inflammatory signals [123]. Additionally, signaling via MAVS leads, as previously mentioned, to a more specific antiviral response [122].

Interestingly, the interplay between metabolism in signaling and the detection of the viral genome has been recently revealed. For example, the role of lactate production in IFN regulation has recently been highlighted [124]. Using a metabolomic approach in a model of human embryonic kidney cells (HEK293) stimulated with poly:IC, the downregulation of glycolysis, illustrated by the decrease in all its intermediates, was demonstrated during the establishment of type I IFN production, dependent on the RLR pathway [124]. In particular, the reduction in glycolysis observed during RLR signaling was associated with a reduction in the mitochondrial localization of hexokinase 2 (HK2), which is essential for hexokinase to exert its full functionality. These data seem to suggest an inverse relationship between glycolysis and antiviral signaling, since the inhibition of glycolysis, either through HK2 knockout or pharmacologically with 2-deoxyglucose inevitably, leads to an increase in TBK1-IRF3 signaling and IFN-β production [124]. To support this idea, the same work highlighted the intrinsic inhibitory function of lactate on RLR activation. The knockdown of the PDHc subunit of pyruvate dehydrogenase led to the impairment of oxidative phosphorylation and the accumulation of lactate. This lactate has been shown to interact with MAVS, limiting its mitochondrial localization, its association with RIG-I, and its aggregation, all of which are essential for signaling in the IFN pathway [124] (Figure 7). Furthermore, while lactate has a direct effect on MAVS, no deficits in the cGAS/STING pathway have been reported in these models [124]. In parallel, and supporting the idea that oxidative phosphorylation, unlike glycolysis, favors MAVS signaling, the production of mitochondrial ROS has been shown to be essential for MAVS oligomerization (Figure 7). The inhibition of mitochondrial ROS formation by MitoQ, an antioxidant that specifically targets mROS, results in a reduction in MAVS oligomerization and a subsequent lower secretion of IFN-β [125]. Mitochondrial ROS are believed to be involved in the peroxidation of mitochondrial membrane lipids, and in turn, the state of the mitochondrial membrane influences MAVS oligomerization [126].

The link between metabolic pathways and viral genome detection has also been demonstrated in the cGAS/STING and TLR pathways (Figure 7). In T cells, the inhibition of mTORC1, a complex involved in the control of metabolism and cell growth, led to a drastic reduction in the production of type I IFN compared to control cells, despite stimulation with cGAMP and a costimulation signal [127]. The importance of mTORC1 on the activity of the cGAS/STING pathway was demonstrated pharmacologically using rapamycin but also in T cells from KO mice for Raptor, a subunit of the mTORC1, leading to a loss of activity of the complex [127]. An investigation of the mechanism through which mTORC1 influences cGAS/STING led to evidence that two downstream factors of mTORC1 (i.e., 4E-BP1 and S6K1) play a role in the antiviral axis, with S6K1 being preponderant [127] (Figure 7). However, it has not been explored whether this effect was due to a variation in metabolism in particular, even though mTORC1 finely regulates T cell metabolism, notably during their activation. On the other hand, it has been previously reported that the specific inhibition of S6K1 by PF-4708671 resulted in exacerbated glycolysis and the inhibition of mitochondrial complex I, illustrating the intimate relationship between S6K1, oxidative phosphorylation, and mitochondrial metabolism [128,129]. These data again support the importance of oxidative phosphorylation in the secretion of type I IFNs. In addition to its role in cellular metabolism, S6K1 interacts directly with STING via its kinase domain to form a tripartite S6K1-STING-TBK1 complex required for IRF3 activation [130]. Similarly, the detection of the endosomal genome by TLRs appears to depend on mTORC1. In mouse pDCs, the stimulation of TLR9 by its ligand in the presence of rapamycin resulted in lower type I IFN production, and this decrease was correlated with a reduction in IRF7 phosphorylation in rapamycin-treated cells [131]. As previously mentioned, the pathway involving S6K downstream of mTORC1 appears to be involved [130]. Endosomal RNA detection also seems to be affected when mTORC1 is inhibited, since the immunization of mice with the yellow fever virus vaccine strain (17D) in the presence of rapamycin results in a decrease in IFN-α/β secretion, this time linked to the downregulation of signaling via TLR7 [131]. Interestingly, the effect of mTORC1 appears to be limited to the cGAS/STING and TLR/MYD88 axes, since the use of poly:IC as an antagonist of RLRs in the presence of rapamycin does not result in a difference in IFN production compared to the control situation [131]. Although these data were obtained in immune cells, it is reasonable to speculate that a similar function could be found in nonimmune cells. Thus, the metabolic context at the time of infection influences the efficiency of interferon secretion.

### 4.2. Metabolic-Dependent Antiviral Response

In fact, a metabolism suited to the secretion of type I and III IFNs is important, since the IFNs thus secreted, via an autocrine activation loop and a paracrine action, lead to the production of antiviral effectors [132]. Because these two types of IFNs are secreted concomitantly, it is difficult to determine their intrinsic effect on the production of interferon-stimulated genes (ISGs). Type I and III IFNs bind to their respective receptors, IFNAR1/2 and IFNLR1/IL-10Rβ, resulting in the phosphorylation and homodimerization of STAT1 or gamma interferon activation factor (GAF). Additionally, it results in the formation of the interferon-stimulated gene factor 3 (ISGF3) complex composed of IRF9 and phosphorylated STAT1 and 2 [132]. ISGF3 then interacts with the interferon-sensitive responsive element (ISRE), while GAF interacts with the gamma interferon activation site (GAS), both leading to the expression of ISGs [132] (Figure 8). ISGs include several antiviral effectors with different functions. While attachment has so far found few limiting factors, apart from heparanase [133], entry is the target of several antiviral effectors, including Myxovirus resistance genes (MX1 and MX2) or proteins of the IFN-inducible transmembrane (IFITM) and tripartite motif (TRIM) families. MX1 and MX2 appear to limit the arrival of viral components in their destination compartment [134,135], while members of the IFITM family, found mainly in late endosomes and lysosomes, inhibit the entry of viruses using these pathways [136,137,138]. However, their mechanism of action remains unclear, with some suggesting that they alter the endosomal acidification essential to the fusion process, while others argue that IFITMs may directly influence the physical properties of endosomal and lysosomal membranes [139].

Among the ISGs inhibiting viral entry, cholesterol-25-hydroxylase (CH25H) has been reported for several viruses, including DENV and ZIKV [140]. CD25H catalyzes the conversion of cholesterol into 25-hydroxy-cholesterol (25HC), an intermediate believed to be involved in the alteration of the physicochemical properties of the membrane and thus, inhibiting the fusion of enveloped viruses [9]. However, 25HC possibly plays a post-entry role as an inhibitor of the sterol synthesis pathway, which is essential for viral progeny [141,142,143,144]. The translation of viral proteins is also a target for antiviral effectors, including proteins of the interferon-induced protein with tetratricopeptide repeat family—IFIT (ISG56/IFIT1, ISG54/IFIT2), the protein kinase R (PKR), or ISG15. PKR and IFIT proteins regulate translation initiation by interacting with the initiation factors eiF2α, eiF3C, or eiF3E [145,146,147,148]. In addition, certain proteins of the IFIT family have the ability to recognize type 0 caps (without 2-O-methylation) or type 1 caps (^m7^GpppA_m_N) leading to the inhibition of translation [149,150]. ISG15 protein, a key player in ISGylation at the post-translational stages of viral and host protein production, has shown multiple functions depending on the *ISGylated* target [151,152,153]. These functions range from IRF3 stabilization and maintaining downstream signaling to increasing the affinity of 4EHP, a negative regulator of translation, for the 5′ cap of messenger RNAs, thus promoting its ability to inhibit translation [152,153]. The establishment of the replicative complex through the deformation of the endoplasmic reticulum (ER) is an essential element in flavivirus replication. This phenomenon is complicated by the expression of interferon α inducible protein 6 (IFI-6), which prevents the ER membrane invaginations required for the replicative complex [154].

In a similar way to type I IFN secretion, we showed that the promotion of oxidative phosphorylation was favorable to the production of antiviral effectors in response to poly:IC stimulation. Indeed, the stimulation of galactose-cultured cells to increase their oxidative phosphorylation resulted in a greater expression of ISG54 and ISG56, two antiviral effectors belonging to the IFIT family [155] (Figure 8). This positive effect on ISG expression was abolished in response to rotenone treatment, undeniably linking the observed effect to mitochondrial chain activity [155]. Recently, we showed that the upregulation of antiviral effector production in galactose-grown cells was actually associated with the anaplerotic feeding of TCA and OXPHOS by L-glutamine [156]. Surprisingly, other colleagues have shown that high glucose, generally propitious to glycolysis, promotes the expression of an essential ISG for the control of Zika virus infection, namely viperin [157]. These results contrast with our own. However, in this work, the expression of other ISGs was not explored [157]. We reconciled these data with our work, showing that cells cultured without L-glutamine, and therefore whose metabolism relies on glycolysis, expressed viperin and OAS1 to a greater extent in response to poly:IC. This observation was indeed related to the glycolytic metabolism of these cells, since the expression of viperin and OAS1 was reduced in the presence of 2-deoxyglucose, an inhibitor of glycolysis [156] (Figure 8). Therefore, these two ISGs appear to be apart when compared with the other effectors we have studied. This reveals the possibility of a dichotomy in ISG expression based on cellular metabolism, with some ISGs having greater expression in a glycolytic context, while others are favored by oxidative phosphorylation and TCA. In addition, the extent to which a virus or a host relies on L-glutamine during infection remains an open and unresolved question in the literature [158]. Is L-glutamine indispensable to the virus during replication, or does the host need it to produce an effective antiviral response [158]? Our work has shown that even without a productive viral infection, L-glutamine is essential for mounting an antiviral response. However, some antiviral effectors do not require L-glutamine to be produced (such as viperin and OAS1). From a dependency point of view, it seems that viruses may have a greater requirement of L-glutamine for their progeny. It is also possible that this bivalence of the antiviral response is an evolutionary adaptation that enables a response to infection regardless of the metabolic context of the infected cell, particularly in the case of virus-induced metabolic reprogramming.

### 4.3. Viral-Induced Metabolic Reprogramming and Associated Immune Evasion

During infection, a virus uses all the cellular machinery and components at its disposal to replicate actively. Among the mechanisms employed by most viruses is the hijacking of the host’s cellular metabolism. This enables the rapid production of large quantities of biomass (nucleotides, lipids, and amino acids) and energy in the form of ATP. This ability to take control of host metabolism has recently been the focus of several in-depth reviews in the literature. Briefly, to meet their need for resources and energy, viruses induce in the infected cell an effect similar to that already identified by Warburg for cancer cells [5], namely the promotion of glycolysis and the diverting of pyruvate toward lactate production, rather than into the TCA, despite the aerobic context—the so-called aerobic glycolysis or Warburg effect. This results in the rapid production of not only energy but also intermediates that may be involved in other metabolic pathways essential for biomass formation, such as the PPP for nucleic acids [4]. Alongside this Warburg-like effect, there is often an increase in glutaminolysis, with glutamine then anaplerotically feeding the TCA, and its intermediates also enabling biomass production [6]. A noteworthy example is the dengue virus, which, through its NS1 and NS3 nonstructural proteins, can increase glycolytic activity and cellular lipogenesis [159,160,161]. ZIKV could also disrupt mitochondrial activity by redirecting the use of glycolysis intermediates toward the PPP. This results in mitochondrial dysfunction that may contribute to pathophysiological processes leading to the congenital defects observed in newborns born to infected mothers [162,163]. The significance of this impact on mitochondrial activity needs to be considered in relation to the cellular model [164]. Additionally, there is a viral-induced action on nutrient receptors, influencing the availability of these nutrients and, necessarily, the metabolic pathways associated with them. The expression of transporters such as GLUT1 can thus be upregulated in various cell types (e.g., epithelial and immune cells), as seen in infections by HIV and the Epstein–Barr virus [165,166,167].

Moreover, since the antiviral response seems to rely on appropriate metabolism to be more effective, targeting metabolism becomes an attractive strategy for immune evasion. Therefore, poxviruses have been shown to produce the F17 protein that has the ability to interact with the Rictor and Raptor subunits of the mTORC1 and mTORC2 complexes, leading to their sequestration and thus the dysregulation of mTOR [168]. As mentioned previously, mTOR is also involved in the regulation of the cGAS/STING-dependent antiviral response via S6K [127]. The sequestration of Rictor and Raptor by F17 thus ultimately leads to a decrease in ISG secretion [168]. Similarly, ZIKV NS4A and NS4B proteins inhibit the AKT-mTOR signaling pathway, even if no exploration of this effect on antiviral gene expression was achieved [169]. Another study showed that the human immunodeficiency virus (HIV) replication relies on the metabolic reprogramming of the host cells. In contrast, the cells of individuals known as “controllers” were refractory to this reprogramming, which could explain the low virus replication in these individuals [170]. Despite antiretroviral treatment, the metabolism of noncontroller LTs is glucose-dependent, whereas controller LTs have a variety of metabolic sources to ensure their survival and function [170]. The controllers’ cells also showed a better effector profile, with a higher expression of IFNB1 and TNF, strengthening the hypothesis of HIV’s ability to hijack the antiviral response by monitoring cell metabolism [170]. Interestingly, the deleterious effect of metabolic reprogramming on the control of viral replication was overcome by treating cells from noncontrollers with interleukin 15. This treatment diversified their cellular metabolism, leading to a significant increase in cellular respiration through enhanced lipolysis [170]. Furthermore, supporting the role of mitochondria in the establishment of the innate immune response, it has been established that mitochondrial elongation associated with the inhibition of dynamin-related protein 1 (DRP1)-mediated fission, via the NS4B protein of the dengue virus, limits the development of the RIG-I-dependent antiviral response in favor of viral replication. While mitochondrial elongation is commonly linked to increased OXPHOS and ROS production, in this context, the alteration in the mitochondria–ER platform appears to be the cause of the inhibition of the antiviral response [171].

## 5. Concluding Remarks

Humanity confronted its first major pandemic of the 21st century through the SARS-CoV-2 pandemic that occurred between 2020 and 2022. While several pathogens served as warnings, gradually marking the onset of the era of emerging viruses, none had, until now, posed such a significant threat to global health and the economy. The work presented here elucidates the influence of metabolism at both the cellular and systemic levels in shaping the antiviral response. This is achieved by attempting to assess the far-reaching impact of host metabolism modulations on the establishment of antiviral responses during infection. The IFN response appears to be a concrete example of immunometabolism applied to antiviral defense. Indeed, the detection of viral molecular patterns depends on an adequate cellular metabolism, just like the production of antiviral effectors, as we have explored. In conclusion, although the viral control of cellular metabolism for its own benefit appears to be a common consequence of infection that affects the antiviral response, our work tends to show that immune evasion through metabolic control appears to have been thwarted by the host. Regardless of the metabolic context, whether glycolytic- or OXPHOS-dependent, the host cell is able to shape its response and produce antiviral effectors: on the one hand, viperin, and on the other, ISGs such as ISG15, 54 or 56. However, as witnessed in the course of several successive epidemics, viruses consistently employ strategies to disrupt the antiviral response during replication, in a frenetic evolutionary race. This crosstalk between virus and immunometabolism, once again, illustrates the red queen effect in infectious processes [172].

## Figures and Tables

**Figure 1 viruses-15-02399-f001:**
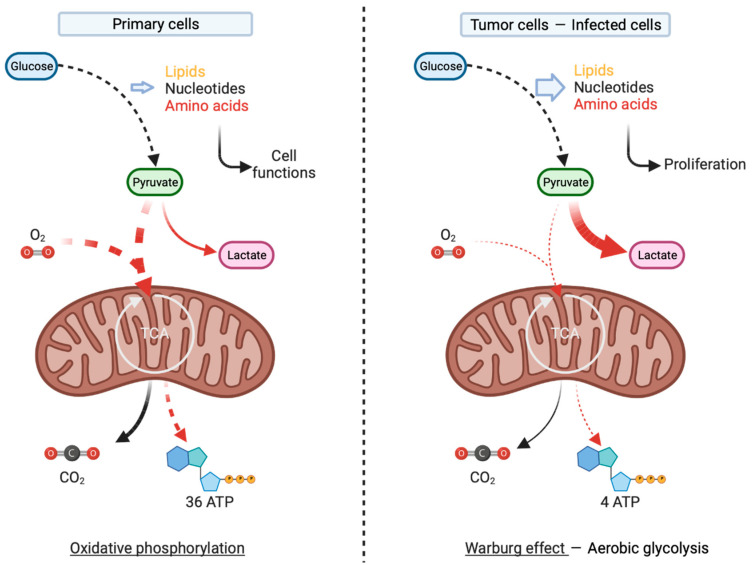
Schematic representation of the Warburg effect. According to this concept developed by Otto Warburg, proliferating cells, including tumor cells, supply their biomass requirements via the adaptation of their metabolism. This adaptation involves the production of lactate from pyruvate even under aerobic conditions and has therefore been established as aerobic glycolysis. The intermediates of glycolysis are then utilized to provide lipids, nucleotides, and amino acids essential for high-rate cellular proliferation.

**Figure 2 viruses-15-02399-f002:**
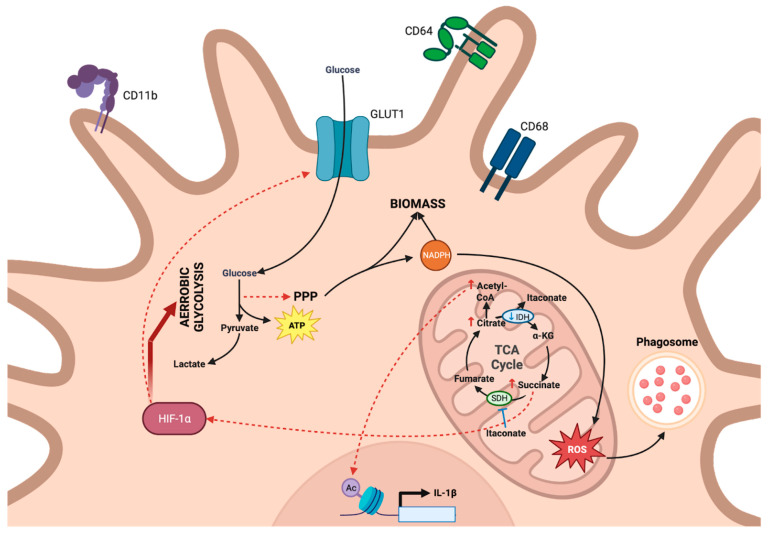
General metabolism involved in proinflammatory M1 macrophage orientation. M1 macrophages develop a glycolytic metabolism enabled by the overexpression of the glucose transporter, GLUT1, in which HIF-1α is involved. The stabilization of this factor is, on the one hand, linked to the inhibition of isocitrate dehydrogenase (IDH-1), leading to citrate accumulation and itaconate synthesis. Accumulated citrate is converted into acetyl-CoA which is used for histone acetylation and thus promotes the transcription of inflammatory genes. On the other hand, itaconate inhibits the succinate dehydrogenase complex (SDH) activity, resulting in succinate accumulation. The succinate thus produced stabilizes HIF-1α. The mitochondrial activity of M1 macrophages contributes to the production of mitochondrial reactive oxygen species, to be used within phagosomes. ATP production therefore relies primarily on aerobic glycolysis.

**Figure 3 viruses-15-02399-f003:**
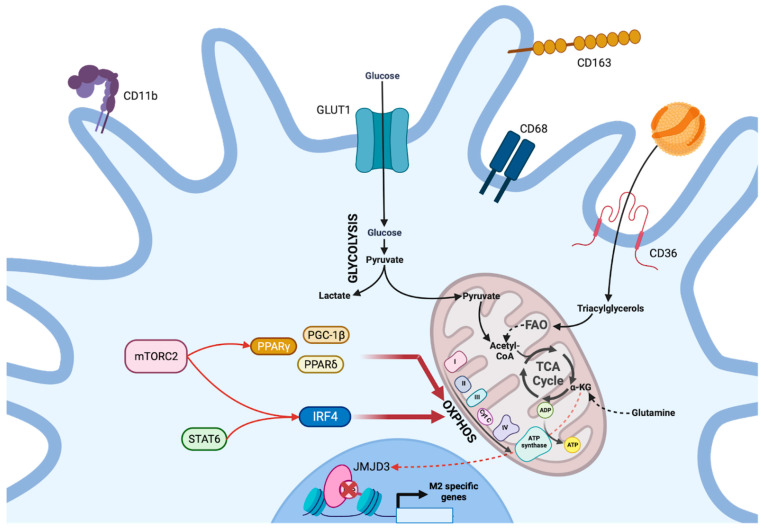
General metabolism involved in anti-inflammatory M2 macrophage orientation. In M2 macrophages, the activation of the PPARδ, PPARγ, PGC-1β, and IRF4 factors enables transcriptional changes directing the cellular metabolism toward OXPHOS. The activation of these factors depends on both STAT6 and mTORC2. In particular, the overexpression of the scavenger CD36 provides lipidic substrates required for fatty acid oxidation. Fatty acid oxidation is essential for ATP production via OXPHOS, rather than for M2 macrophage functions. On the other hand, glutamine, which supplies the Krebs cycle, is essential since it orchestrates M2 polarization through the activation of JMJD3 demethylases.

**Figure 4 viruses-15-02399-f004:**
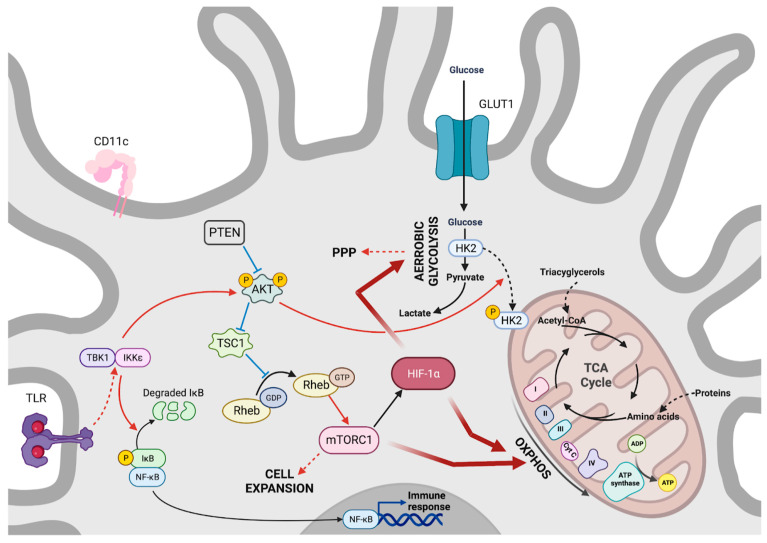
Dendritic cell metabolism. Maintenance of dendritic cells’ quiescence depends on an overriding activity of mTORC1. PTEN and TSC1, negative regulators of mTOR, limit dendritic cell growth. mTORC1 orientates quiescent cells toward OXPHOS utilization, with protein catabolism and triacylglycerol providing amino acids and fatty acids to supply ATP production via the Krebs cycle. Activated dendritic cells take up glucose as the main fuel for a metabolism geared toward aerobic glycolysis. The activation of mTORC1and HIF-1α, in this case, is enabled by TBK1 and IKKε activity on AKT downstream of dendritic cells PRR.

**Figure 5 viruses-15-02399-f005:**
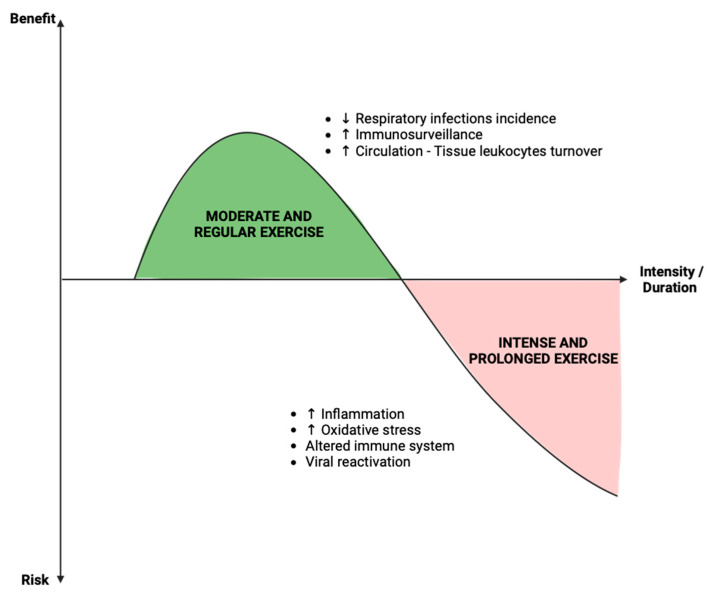
Benefice and risk of physical activity in the context of infection. The role of physical activity on immunity in the case of viral infection remains a subject of debate in the literature. The data available to date seem to point toward the description of a hormesis phenomenon. Physical activity has a beneficial effect, illustrated by a reduction in the incidence of infection and an increase in immune surveillance, in the case of regular and moderated activity. On the other hand, when the “dose” of exercise becomes intense and prolonged, deleterious effects and a context conducive to infection are found, namely an increase in inflammation and oxidative stress, and alteration in the immune system.

**Figure 6 viruses-15-02399-f006:**
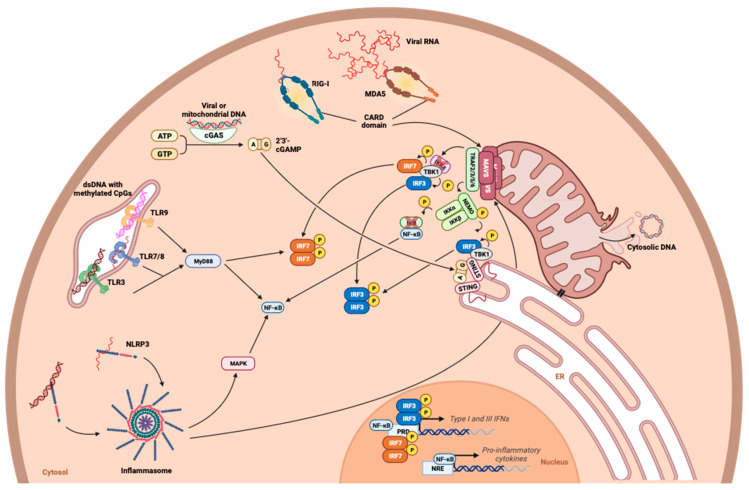
Molecular mechanisms leading to the detection of viral genome. The detection of RNA or DNA from viral origin is dependent on several pathways acting in concert to establish a rapid and efficient antiviral response. These include the activation of RIG-I-like (RLR) receptors and Toll-like (TLR) receptors, but also of the inflammasome via NLRP3. In addition, cytosolic viral DNA or released DNA resulting from mitochondrial damage leads to detection through the cGAS/STING pathway. All these warning signals are integrated and lead to the activation of transcription factors common to these detection pathways, namely IRF3, IRF7, and NF-κB. These transcription factors interact with their promoters to enable the expression of type I and III interferons, on the one hand, and proinflammatory cytokines on the other.

**Figure 7 viruses-15-02399-f007:**
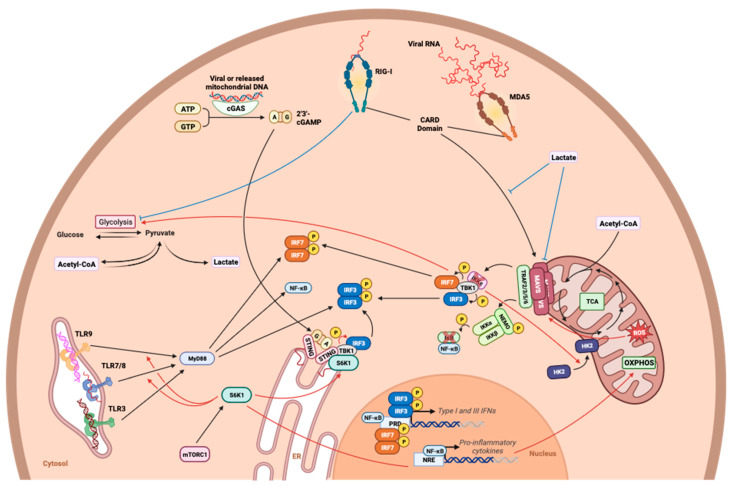
Involvement of metabolism in viral genome detection. The study of the influence of metabolic pathways on the detection of viral pathogens reveals the intricate relationship between metabolic players and viral genome detection. The mTORC1 pathway and S6K, which is a part of this major signaling pathway, have shown their role in promoting cGAS/STING and TLR-dependent detection pathways. On the other hand, aerobic glycolysis and its final metabolite, lactate, interact negatively with the RLR-dependent detection pathway, notably by inhibiting MAVS.

**Figure 8 viruses-15-02399-f008:**
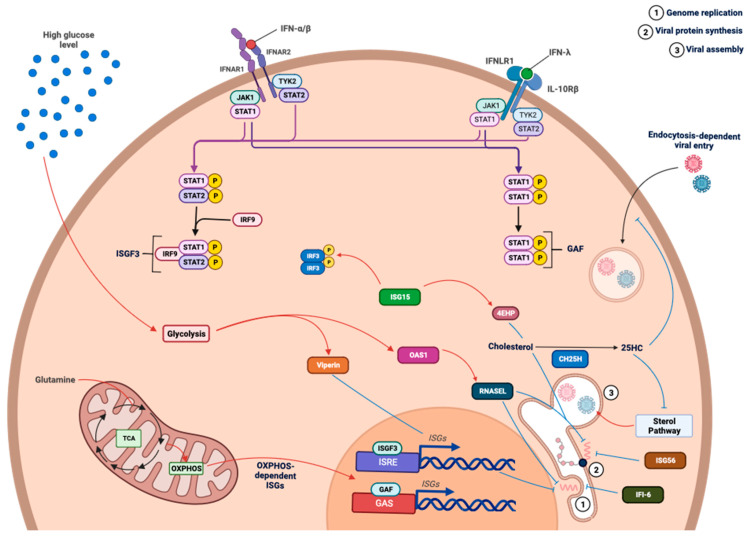
Antiviral effectors and cell metabolism interactions. The binding of type I and III interferons (IFNs) to their respective receptors results, through the JAK/STAT pathway, in the nuclear translocation of ISGF3 and GAF factors. These factors interact with their respective promoters, ISRE and GAS, leading to the expression of interferon-stimulated genes (ISGs), which act as effectors in the antiviral response. During viral infection, several of these ISGs have exhibited antiviral activity, whether during virus entry, replication, or assembly. Other ISGs, such as ISG15, play a role in promoting the antiviral response by enhancing the activity of certain factors involved in this response. Metabolism and the antiviral response are intricately linked, particularly in the expression of antiviral effectors. Oxidative phosphorylation, for instance, tends to enhance the expression of certain interferon-stimulated genes (ISGs) like ISG15, ISG54, or ISG56. However, a duality in this response appears to exist, as some genes encoding antiviral effectors are more expressed during aerobic glycolysis, suggesting they might serve as a backup mechanism in the event of metabolic reprogramming during infection.

## Data Availability

Data sharing is not applicable.

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
