# Peer review of "The Influence of Metabolism on Immune Response: A Journey to Understand Immunometabolism in the Context of Viral Infection"

_viruses, 2023, doi:10.3390/v15122399_

Round 1

Reviewer 1 Report

Comments and Suggestions for Authors

The manuscript submitted by Safadi et al describe the metabolic regulation of immune response. In the manuscript, author present a review of current insight on the immunometabolism and antiviral response during viral infections. Also, the examples of immune cells activation intricately regulated by metabolic processes are explored. The review in the manuscript is quite complete and has a certain degree of innovation, which provide the guidance between metabolism in antiviral defense and interferon response.

 There are some flaws in the manuscript.

Some subheadings are too brief, such as Line 98 “2. Immunometabolism”, Line 297 “i. Diabetes”, Line 371 “ii. Obesity” and line 452 “b. Exercise”. These subheadings should briefly summarize the relevant content.

Line 762, the content of “Concluding remarks” is not systematic, which should be systematically summarized to be the meaningful insights.

Reviewer 2 Report

Comments and Suggestions for Authors

The authors have comprehensively reviewed the role of metabolism in regulating the immune response.  My only concern is that the review is primarily concerned with the general role of metabolism and only uses approximately one fourth of the manuscript addressing viral infections.  Three fourths of the manuscript is concerned with metabolic regulation in cancer exercise etc.  While there is inherently nothing wrong with being exhaustive it makes the title misleading since the emphasis is really not on the effects in viral infections.  Perhaps the easiest way to handle this criticism would be to revise the title to better represent the focus.  There is a typo in line 126. 
